# Spatial Distribution of PM_2.5_ Mass and Number Concentrations in Paris (France) from the Pollutrack Network of Mobile Sensors during 2018–2022

**DOI:** 10.3390/s23208560

**Published:** 2023-10-18

**Authors:** Jean-Baptiste Renard, Eric Poincelet, Isabella Annesi-Maesano, Jérémy Surcin

**Affiliations:** 1LPC2E-CNRS, 3A Avenue de la Recherche Scientifique, CEDEX 2, F-45071 Orléans, France; 2Pollutrack, 5 rue Lespagnol, F-75020 Paris, France; e.poincelet@gmail.com (E.P.); jeremysurcin.pollutrack@protonmail.com (J.S.); 3Institute Desbrest of Epidemiology and Public Health, Allergic and Respiratory Diseases Department, Montpellier University Hospital and INSERM, Montpellier, IDESP IURC, 641 Avenue du Doyen Gaston Giraud, F-34093 Montpellier, France; isabella.annesi-maesano@inserm.fr

**Keywords:** PM2.5, urban pollution, Paris (France), maps

## Abstract

The presence of particulate matter smaller than 2.5 µm in diameter (PM2.5) in ambient air has a direct pejorative effect on human health. It is thus necessary to monitor the urban PM2.5 values with high spatial resolution to better evaluate the different exposure levels that the population encounters daily. The Pollutrack network of optical mobile particle counters on the roofs of hundreds of vehicles in Paris was used to produce maps with a 1 km^2^ resolution (108 squares to cover the Paris surface). The study was conducted during the 2018–2022 period, showing temporal variability due to different weather conditions. When averaging all the data, the highest air pollution was found along the Paris motorway ring. Also, the mean mass concentrations of PM2.5 pollution increased from southwest to northeast, due to the typology of the city, with the presence of canyon streets, and perhaps due to the production of secondary aerosols during the transport of airborne pollutants by the dominant winds. The number of days above the new daily threshold of 15 µg.m^−3^ recommended by the WHO in September 2021 varies from 3.5 to 7 months per year depending on the location in Paris. Pollutrack sensors also provide the number concentrations for particles greater than 0.5 µm. Using number concentrations of very fine particles instead of mass concentrations corresponding to the dry residue of PM2.5 is more representative of the pollutants citizens actually inhale. Some recommendations for the calibration of the sensors used to provide such number concentrations are given. Finally, the consequences of such pollution on human health are discussed.

## 1. Introduction

Particulate matter (PM) pollution has a direct effect on human health [1,2], in particular in urban conditions, where the PM concentrations increase due to the combination of meteorological conditions, local sources, and imported pollutants, and where the impact of their abatement has been estimated [3]. The smallest particles are by far the most dangerous ones since they can penetrate deeper in the body [4]. They are found in various organs [5,6,7], responsible for cardiopulmonary mortality and cancers [8], and even related to peaks in COVID-19 mortality [9,10,11].

PM2.5 refers to particles with an aerodynamic diameter ≤ 2.5 µm. In urban conditions [12], these particles can be directly produced by various anthropogenic sources, mainly traffic, building activities, and wood and oil-fired boiler heating (primary PM), but also from complex chemical reactions involving sunlight, anthropogenic ammonium, nitrous oxides, and sulfur oxides (secondary PM). The recent recommendations from the World Health Organization (WHO) in September 2021 for PM2.5 mass concentrations include a mean annual value below 5 µg.m^−3^ and a 24 h value below 15 µg.m^−3^, a ceiling limit that should not be exceeded for more than 4 days per year [13]. It is thus of importance to monitor the spatial and temporal evolution of the concentrations of PM2.5 to evaluate the exposure of urban populations to polluted ambient air.

Paris (France) is a city of 2.2 million citizens, surrounded by a suburb of 10 million inhabitants. The Paris region is characterized by relatively few polluting industries, but intense traffic and agricultural activities close to the Paris boundaries. Paris also has the highest population density in the Western world and attracts nearly 50 million visitors per year. The region’s urban background conditions are dominated by long-distance transportation of pollutants, accounting for up to 70% of the PM2.5 mass concentration [14,15,16,17,18,19,20]. On the other hand, Paris is a very compact city (105 km^2^), encircled by a motorway ring (called “périphérique”), with dense traffic of up to 250,000 vehicles per day. Local sources of primary particles are dominated by car traffic emissions (exhaust from diesels, tires, breaks, dust from road material) and by residential heating in winter [21]. As a consequence, the spatial distribution of particle concentrations can vary significantly depending on the transport of air mass driven by the urban topography [22]. It is thus necessary to study the variation in the exposure of citizens to pollution at the local level.

The PM2.5 mass concentration is continuously monitored using regulatory air quality networks that provide reference mass concentrations. In the Paris region, these measurements are recorded by Airparif (accessed on 27 April 2023 at https://www.airparif.asso.fr/). Six PM2.5 stations are now available in Paris, with two of them in proximity to the motorway ring but none of them in the south or northeast of the city. The mass concentration technique provides a good estimate of the presence of the largest particles but cannot provide estimates of the contribution of the submicron ones alone [23]. Thus, measurements of number concentrations as a complement to mass concentrations may be necessary to better understand the distribution and dynamics of urban PM.

Mobile measurement stations utilizing counting instruments can be used to map the local content of PM2.5 at the street, and therefore pedestrian and biker breathing, level. This is the main objective of the Pollutrack network using low-cost but accurate mobile instruments. The first PM2.5 hyperlocal maps based on actual measurements rather than through modeling were presented by Renard and Marchand [23], showing the first analysis of the local variability in PM2.5 content in Paris for the year 2020. The aim of the present study was to deepen the analysis of the mean spatial variability in PM2.5 across Paris and its motorway ring over the 2018–2022 period, using 1 km^2^ resolution maps for the PM2.5 mass concentrations levels and for the number of days above the WHO recommendations. We present the measurement strategy, the instrument accuracy determined from comparison with reference measurements, the temporal variability, the 1 km^2^ resolution maps of the spatial variability, the corresponding number concentrations, and a discussion of the meaning of the results in terms of human health.

## 2. PM2.5 Measurements with the Optical Pollutrack Sensors

### 2.1. Strategy of Measurements

The core of a Pollutrack sensor consists of a compact optical particle counter, where the particles are injected inside an optical chamber to cross a laser beam. The size of the individual particle is determined from the intensity of the light scattered by individual particles. The instrument provides the number concentration of particles in the 0.5–1 µm, 1–2.5, 2.5–5, and 5–10 µm size ranges each second. The counts are converted into PM2.5 and PM10 mass concentrations using internal calibrations. Since this conversion is sensitive to the relative humidity, which can affect the mass density of the particles [24], a corrective factor is applied during high-humidity episodes (these values are retrieved in real time from local weather stations). In general, such small PM sensors may be less accurate than more expensive and heavier number concentration counters and mass concentration instruments; they can, however, provide good indications of the PM mass concentrations [25].

The sensors have been mounted on the roofs of professional fleets of vehicles, most of them being electric [23]. In Paris, three partners have fleets circulating in almost all streets of Paris and its inner suburbs (Enedis, the national public electricity distributor; Geopost/DPD group, the largest European parcel delivery service; Marcel Cab, contributing a fleet of compact electric taxis). Up to around 500 cars were equipped with Pollutrack sensors during the 2018–2022 period. The sensors are oriented in the opposite direction of car motion, guaranteeing a more stable airflow and therefore the optimized use of the inlet system. The device considers the relative speed between the inlet and the wind for particles of a few µm and for a relative speed of about 40 km.h^−1^. The mobile Pollutrack system was mainly designed for the estimation of PM2.5 mass concentration, widely considered as the most dangerous of all urban airborne pollutants. Finally, fixed stations with the same sensors have been installed at different locations such as vehicle depots to perform daily quality control of all mobile sensors and thus eliminate all erroneous measurements.

A statistical approach is used by the Pollutrack network to increase measurement accuracy. Instead of considering the results from a given instrument and its path of travel, Paris is divided into small square parcels, where all the measurements are averaged to provide a mean mass concentration value in each parcel. The size of the parcel and the duration of integration can be chosen depending on the number of measurements. In the following, we consider a square with a 1 km side length, and an integration time of 24 h to be in line with the normative law regulation regarding daily values.

### 2.2. Validation of the Sensors

A first attempt to validate the Pollutrack sensors was conducted [23], showing an uncertainty of about 5 µg.m^−3^. Improvements in the data retrieval have been performed since this first publication. Thus, new sessions of intercomparison with 9 Pollutrack sensors in parallel and 3 PM2.5 reference stations (BAM 1020 from Metone) of the Airparif network were conducted in 2018 for the City Hall authorities of Paris (Figure 1). Three different sites were considered: traffic conditions in the motorway ring (“Peripherique Est”) from 26 April to 15 October 2022, the urban background (garden in the center of Paris “Les Halles”) from 13 July to 15 October 2022, and rural conditions (“Rambouillet”) from 10 July to 15 October 2022 The instruments were mounted close together, ensuring the sampling of similar air masses. The analysis showed that first the mean difference between the various copies of the Pollutrack sensors during the three sessions of measurements was 0.4 ± 0.3 µg.m^−3^. This confirmed the good reproducibility of the sensors, an essential feature for the original goal of the Pollutrack network, namely, relative measurements between districts and streets, however also backed by good absolute values. 

Then, the daily averaged values of the PM2.5 were calculated both for Airparif reference stations and Pollutrack sensors at the three locations. Figure 2 presents an example of the intercomparison at the Peripherique Est station (time evolution and correlation plot). The statistics for the three sessions of measurements are given in Table 1, although the range of the pollution values is not large, and the presence of few isolated values for the highest mass concentrations can bias the retrieval of the slope values on the correlation plots.

To confirm these results, three other sessions of intercomparison with Pollutrack sensors and PM2.5 reference microbalance were conducted at 3 different locations in Dublin (Ireland), which is a port city, for the entire year of 2022. Similar results to those for the Paris region were obtained in terms of the measurements statistics (Table 2). 

The daily difference between the Pollutrack and reference measurements were calculated, then the mean difference was calculated. For all these intercomparison sessions, a mean value of ~3 µg.m^−3^ was found. It could be concluded that the values of individual Pollutrack sensors had a mean uncertainty of 3 µg.m^−3^. Increasing the number of sensors and measurements at the same location, as performed with the mobile network, should reduce this uncertainty; thus, this value can be considered an upper limit.

## 3. PM2.5 Maps 

### 3.1. Maps Retrieval

The total surface of Paris city was divided into squares with a 1 km side length to ensure hundreds of Pollutrack measurements per day and per square. Then, these data were integrated over 1 day to provide daily values per square for a 5-year period. This was statistically relevant since the measurements were randomly distributed along the daytime and since the cars covered all the main streets in Paris. These 108 parcels provided a strong improvement in spatial sampling compared with the 3 to 6 PM2.5 stations of the Airparif air quality monitoring network. It was then possible to study the temporal trend of the PM2.5 pollutants per 1 km^2^ square.

Renard et al. [22] used a different approach using the 2020 Pollutrack data to produce examples of daily PM2.5 maps with a resolution of 100 m along main roads only. This approach is specifically useful to detect hot spots but is less suitable for identifying spatial global trends. Thus, the approaches are complementary.

The 108 Pollutrack daily values were averaged to produce one global value per day that could be compared to the daily mean of the fixed Airparif station values to verify the consistency of the two sets of data. As shown in the correlation plot in Figure 3, the agreement between the two sets is very good, with a mean difference of 0.1 ± 3.5 µg.m^−3^. It can be concluded, first, that the mean of the few fixed monitoring stations can provide a good estimate of the mean pollution in Paris, except for a few measurements where the values are too high due to local and short-duration pollution events, such as the isolated 70 µg.m^−3^ Airparif value in Figure 2. Second, we concluded that no systematic bias was present in the Pollutrack measurements. Finally, the data were also averaged for each 1 km^2^ square over the 5-year period of measurements to establish the mean spatial trend in the PM2.5 pollution levels in Paris.

### 3.2. Temporal Trend

The “high season” of PM pollution (meaning the period of the year when the highest mass concentration values are measured) occurs between autumn and the beginning of spring, corresponding to the combination of a large number of sources, essentially traffic, heating, and agricultural activities [14,21]. The pollution increases steadily under anticyclonic conditions, with the occasional presence of an inversion layer that maintains the pollution close to the ground. All the time evolution curves for the 108 squares with a 1 km side length are plotted in Figure 4. The curves show the strong PM2.5 content variability from one location to another and the trend in PM2.5 content over the 5-year period. The strongest high season occurred during winter 2018–2019, while the lowest high season values occurred during winter 2019–2020. The lower 2020 values were due to both the natural yearly weather variability and the lowering of anthropogenic activities linked to the COVID-19 pandemic [26]. The new WHO recommendation values are represented by dashed lines in Figure 4. It can be seen that almost no locations in Paris were below the annual 5 µg.m^−3^ value. Also, the daily ceiling of 15 µg.m^−3^ was exceeded for a large number of days at almost all locations. Table 3 provides the yearly lowest mean, mean, and highest mean values of PM2.5 mass concentrations from the 108 (1 km^2^) squares.

### 3.3. Spatial Trend

All the Pollutrack daily data were averaged for the 2018–2022 period to retrieve the mean mass concentrations values for each 1 km^2^ square (Figure 5). Three main features can be distinguished:-The mean PM2.5 pollution was higher in the northeast of Paris than in the south-west, with a ratio of 1.5.-Not surprisingly, the value was almost always higher along the motorway ring than in its surroundings.-Four permanent hot spots were present in the high-density motorway connections to the motorway ring.

The features result from the calculation of the mean over 5 years. Obviously, the mean yearly values differed from one year to another (Table 3), although the main features globally remained the same. The ratio between the highest and lowest mean values remained in the 1.4–1.8 range, indicating that the local traffic was indeed the main source of pollution at the Paris motorway ring locations. At distances larger than about 1 km, the effect of the motorway pollution seemed to disappear.

The mean value is not a sufficient parameter for evaluating the consequences of pollution exposure on human health. The number of days above the 15 µg.m^−3^ WHO recommended 24 h ceiling value must be also considered. Figure 6 presents the mean number of days above this threshold for the 2018–2022 period. As for the mean mass concentrations, a southwest–northeast dichotomy was found, and the highest values correspond, expectedly, to the Paris motorway ring. On the other hand, the mean ratio between the lowest and highest number of days was 2.1 (from ~3.5 to ~7 months) and in the 1.8 – 4.3 range when considering the individual yearly values (Table 4). Then, the variability of the number of days exceeding this threefold more accurately represents the spatial and temporal heterogeneity in the citizen exposure to PM2.5 pollution.

## 4. Discussion

### 4.1. Origins of Spatial Heterogeneity

The origin of the PM2.5 pollution near the Paris motorway ring can be easily related to the local sources of traffic. On the other hand, the southwest–northeast dichotomy could have two origins. The first one is related to the topology of Paris. The north-center, north, and east of Paris are dominated by narrow streets in which the canyon effect may occur [27]. Such streets are not laterally ventilated, and PM2.5 may accumulate, similar to the snowdrift effect. The second one could be related to the wind direction. Winds come from the southwest direction during half the year. One possibility could be the formation of secondary aerosols above Paris during some periods in the year, for example, when ammonium coming from the large agricultural zones in the west and south of Paris react with the local NOx emissions during their transport.

The origin of aerosols is complex in Paris [20,28], as with other major cities. New complex modeling work on aerosols transport should be conducted, among others, in view of the relationship between COVID-19 mortality or other respiratory disorders that occur owing to PM2.5 spikes [11], and this modeling should be applied to all the streets of Paris to better understand the role of natural ventilation in the local dissipation of pollution. 

### 4.2. Number Concentrations

All these analyses are based on mass concentration, which is just one parameter used to characterize aerosols. Mass concentration is more sensitive to the presence of large particles, which are intrinsically the heaviest ones. Other parameters such as the number concentration and the size distribution of the particles must be considered. They can provide some valuable information about the origin of the particles, their mode of production, and their transport scheme [29,30]. Also, the number concentration provides better information on what is actually inhaled by a person, targeting the concentration of particles that enter deeper in the body with respect to the particle size. 

In Figure 7, a correlation plot is shown between Pollutrack mass concentrations and the total number concentrations of particles > 0.5 µm. Different total number concentrations with a dispersion of up to a factor of two, characteristic of different size distributions, can produce the same mass concentration values, thus hiding the actual health impact of fine and very fine particles. The number concentrations corresponding to a given range of mass concentrations were averaged over the 5-year period. The evolution of these mean values followed a linear trend. To establish a map of the number concentrations (Figure 8) that can be compared with that of the mass concentrations (Figure 5), a color scale was established using a multiplicative coefficient calculated from this linear trend, except for the number concentration upper limit, which needed to be increased to include the highest values. 

Some differences can be found between Figure 5 and Figure 8. The dichotomy is less pronounced in the number concentrations map, and the location of higher pollution levels, motorway ring excluded, is more concentrated in the northeast of Paris. This feature is in agreement with the hot spots previously detected [23]. The difference between these two maps is due to how the submicron particles are considered. They are not accounted for so much in the mass concentration calculations, while they dominate the number concentration measurements. These results suggest that, in the future, official recommendations and general public alerts should be based on PM2.5 and not PM10, as is still the case in Europe, and PM2.5 levels should be based on number concentration instead of mass concentration.

Nevertheless, some difficulties exist with the approach. The main one is the definition of the size that depends on the method of measurements (equivalent diameter, aerodynamic diameter, electric mobility diameter, etc.). In particular for irregular-shaped particles, all these diameters can significantly differ, so subsequently need sophisticated calculations to reconcile them [30]. Also, most of the optical counters are sensitive to the refractive index of the particles [31], which can distort size attributions and lead to significant bias. Finally, conventional optical counters cannot detect particles smaller than about 200 nm, although the maximum number concentration of urban pollution particles is in the 100 nm range for urban pollution. Thus, they underestimate the number of submicron particles, although providing an acceptable indicator of their presence. 

The number concentrations values presented here are directly related to the Pollutrack sensor size calibration and may differ when using another optical aerosol counter, although the relative concentration variations should remain similar. One possibility in the future could be to use dedicated particles to intercalibrate the various instruments. Of course, submicron latex beads may be used, as applied at present in all the optical aerosols counters [32], but since natural particles are not perfect solid spheres, irregular grains must be also considered [31]. As an example, SiC powders of 5 µm or larger, which are sized-calibrated powders used for polishing purposes, can be used, as well as complex calibration procedures with spherical and nonspherical particles [33].

### 4.3. Towards a Better Analysis of the Effects of PM Pollution on Health 

The 1 km^2^ resolution map shows that no Paris location is in line with the new WHO recommendation, even for daily ceiling values. People living and/or working close to the motorway ring or in canyon streets can inhale on average ~twice as many particles (>0.5 µm) than in the better-preserved southwest districts of Paris.

Various systematic reviews revealed that PM2.5 is associated with increased short-term and long-term mortality and morbidity due to respiratory, cardiovascular, and cerebrovascular disorders, and diabetes [34]. The high level of PM2.5 pollution can explain the high mortality that occurred in Paris during the COVID-19 pandemic, which was one of the highest values in western Europe [10,11]. It would have been of interest to tentatively correlate the spatial trend in COVID-19 mortality with the spatial trend in PM2.5 pollution in Paris, but unfortunately, such medical data are not currently available. Similarly, to better understand the effect of PM2.5 on severe respiratory pathologies, future analysis will be necessary using mortality data in parallel with PM2.5 data, but again, such data are not currently available for that purpose. 

The analysis presented here should be conducted for other polluted cities around the world. Most of the studies (~69%) on the health effect of PM pollution were carried out in cities of high-income countries, despite the fact that PM2.5 concentrations are higher in low- and middle-income countries [35], where severe sanitary effects should be expected. Finally, chemical components and sources apportionment of PM2.5 to the associated toxicity have been poorly documented. Thus, a better determination of PM origin should be conducted in parallel with PM content determination and health analysis, using instruments that can provide the type and composition of the PM (as conducted during pollution events in Paris [14]), and the access to more local sanitary data should be increased.

## 5. Conclusions

The Pollutrack mobile sensors network allowed the local mapping of PM2.5 in Paris, based on real measurements, with a 1 km^2^ spatial resolution. When averaging all the data from 2018 to 2022, a trend was found, with higher pollution values in the northeast than in the southwest of Paris, with the highest values expectedly found along the motorway ring around Paris. When compared with regulatory monitoring based on PM10 (and even PM2.5) weighing mass concentrations, the results using the number concentration of PM seem to be better suited for evaluating what the citizens really inhale in terms of deleterious particles with a negative health impact. The number of particles per cm^3^ increases when the particle size decreases, and the maximum concentrations occurs at around 100 nm for urban pollution. Nevertheless, the mass concentrations of such submicron particles remain weak compared with those of the largest one, and thus do not always strongly contribute to the final PM10 and PM2.5 mass concentrations. Thus, the amount of the smallest particles cannot be determined when using mass concentration instruments, and pollution events dominated by such dangerous tiny particles could be undetected when considered in conjunction with the regulatory pollution thresholds.

Only the mean values were here considered, but Pollutrack maps are produced daily and will soon become accessible to the general public with Paris authorities’ authorization. Such daily maps can help with identifying the PM2.5 pollution hot spots due to, for instance, specific events like building sites or important traffic congestion. Additionally, the maps may also identify canyon streets that may be harmful to asthmatic people, resulting in the identification of opportunities for remedial actions such as pedestrianization and proper ventilation. Furthermore, the Pollutrack data and maps could be used to monitor the effect of new regulations that will limit the traffic within Paris.

Beyond Paris, where Pollutrack was originally conceived at COP21, this mobile network is now deployed in over thirty capitals and major cities (including London, Dublin, Brussels, Geneva, Zurich, Rotterdam, and Hamburg) across Europe, which is the continent most highly exposed to diesel pollution. These data should be used in particular by local authorities to better understand the variability in PM2.5 in their cities and to improve urban air quality, with the objective of reducing the mortality due to PM2.5 pollution. 

## Figures and Tables

**Figure 1 sensors-23-08560-f001:**
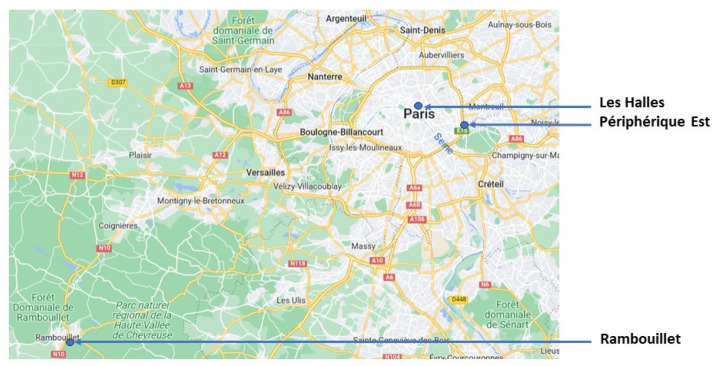
Location of the measurements for the intercomparison between Pollutrack sensors and air quality stations of Airparif (map from Google Maps).

**Figure 2 sensors-23-08560-f002:**
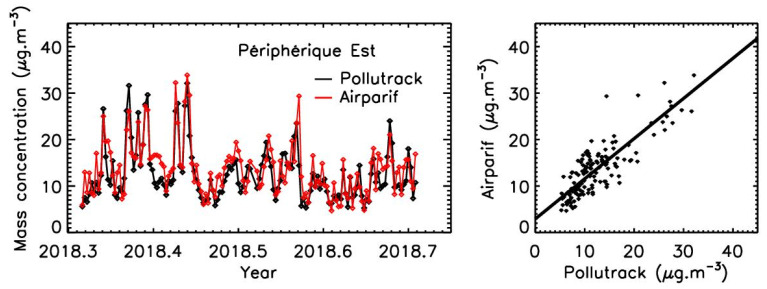
Example of intercomparison between the PM2.5 reference station of the Airparif air quality network (daily average) under traffic conditions at the motorway ring of Paris Périphérique Est. Left: Daily evolution of the mass concentration. Right: Correlation plot.

**Figure 3 sensors-23-08560-f003:**
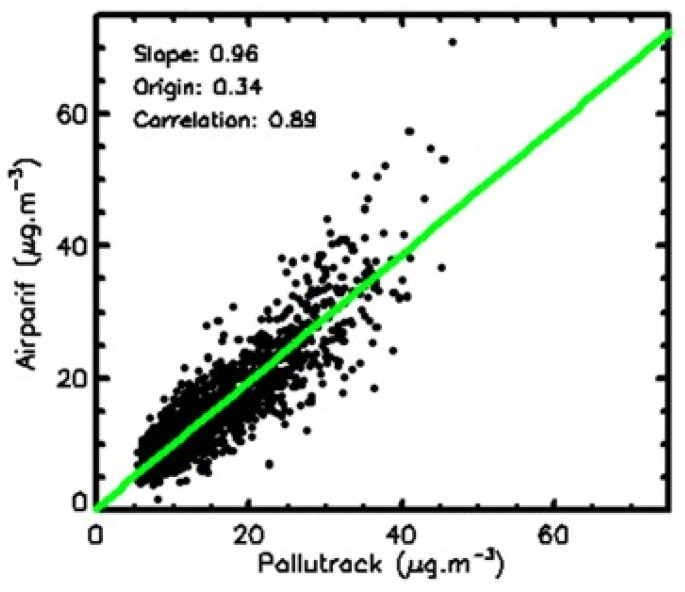
Correlation plot between daily mean PM2.5 Pollutrack and Airparif measurements in Paris. The green line represents the linear fit obtained by a linear regression.

**Figure 4 sensors-23-08560-f004:**
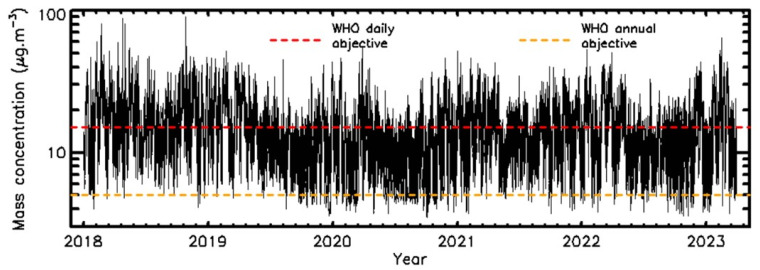
Time evolution of the PM2.5 mass concentrations, plotting the 108 1 km^2^ squares of Pollutrack measurements, compared with WHO recommendations.

**Figure 5 sensors-23-08560-f005:**
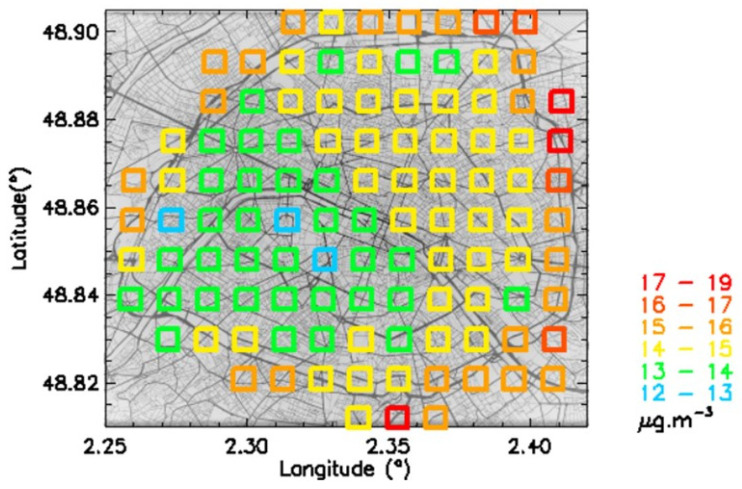
Map of mean PM2.5 mass concentrations for the 2018–2022 period.

**Figure 6 sensors-23-08560-f006:**
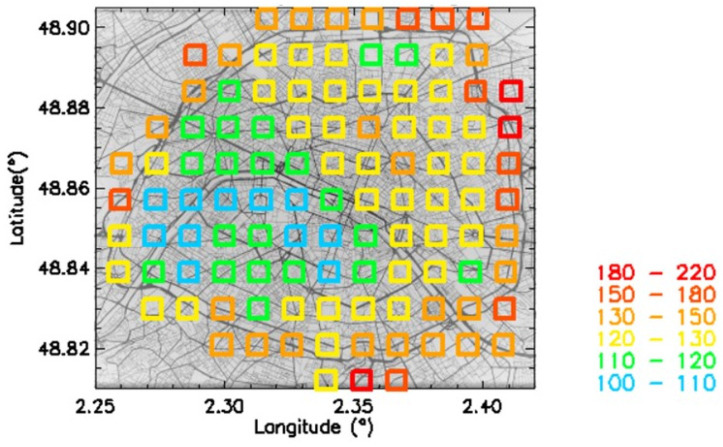
Map of mean number of days above the 15 µg.m^−3^ threshold for the 2018–2022 period.

**Figure 7 sensors-23-08560-f007:**
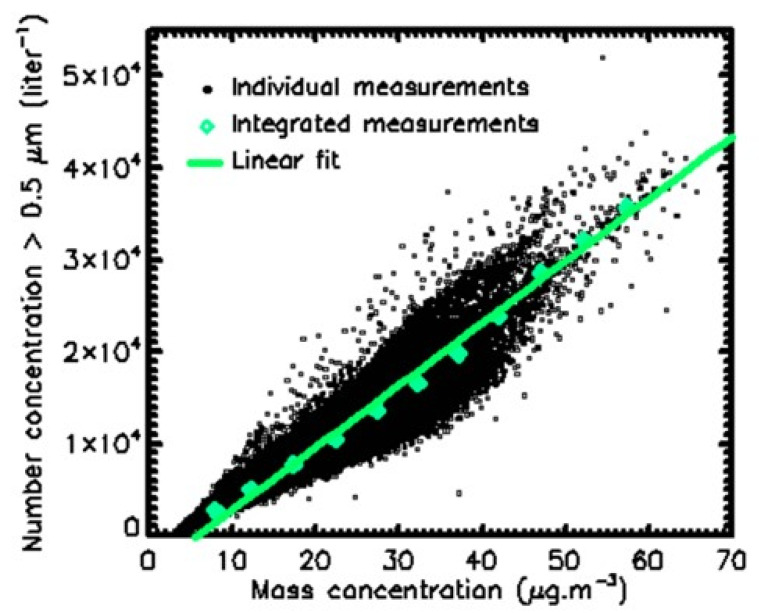
Correlation plot between daily mass concentrations and number concentrations for the 180 1 km squares for the 2018–2022 period. The green line represents the linear fit obtained by a linear regression.

**Figure 8 sensors-23-08560-f008:**
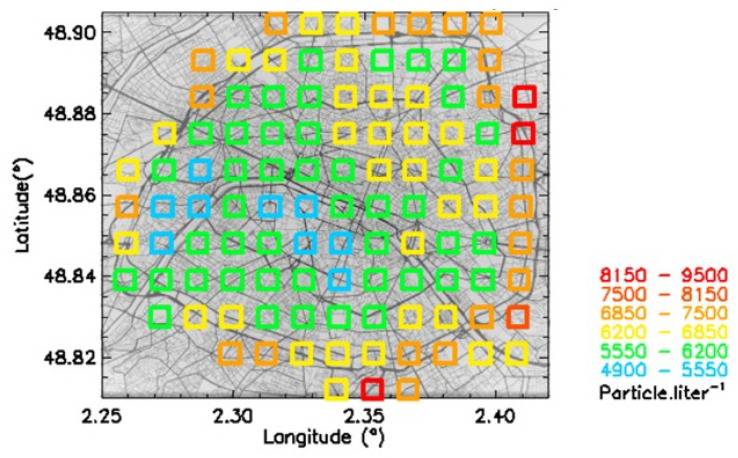
Map of mean PM2.5 number concentrations for the 2018–2022 period.

**Table 1 sensors-23-08560-t001:** Statistics of the comparison between Pollutrack measurement and air quality Airparif PM2.5 measurements at 3 different locations in the Paris region.

Station	Périphérique Est	Les Halles	Rambouillet
Pollution range (µg.m^−3^)	5–34	5–26	4–18
Correlation	0.83	0.85	0.83
Slope	0.86	1.01	0.93
Value at origin (µg.m^−3^)	2.9	0.9	−0.4
Standard deviation (µg.m^−3^)	3.2	2.4	1.8

**Table 2 sensors-23-08560-t002:** Statistics of the comparison between Pollutrack measurement and Dublin air quality network measurements from the 3 different locations in Dublin (Ireland).

Pollution Range (µg.m^−3^)	5–50
Correlation range	0.88–0.90
Slope range	0.87–1.19
Range of value at origin (µg.m^−3^)	−1.4–0.0
Standard deviation range (µg.m^−3^)	1.5–3.1

**Table 3 sensors-23-08560-t003:** Lowest yearly mean, yearly mean, and highest yearly mean mass concentration values in the 108 1 km^2^ squares.

Period	Lowest Mean Mass-Concentration (µg.m^−3^)	Mean Mass-Concentration (µg.m^−3^)	Highest Mean Mass-Concentration (µg.m^−3^)
2018–2022	12.6	14.4	18.7
2018	15.4	17.1	21.6
2019	12.7	14.3	18.4
2020	9.3	12.1	16.7
2021	12.1	14.6	19.1
2022	11.5	14.0	18.8

**Table 4 sensors-23-08560-t004:** Lowest, mean, and highest number of days per year above the 15 µg.m^−3^ WHO recommended 24 h threshold for the 1 km^2^ squares.

Period	Lowest Number of Days	Mean Number of Days	Highest Number of Days
2018–2022	102	129	218
2018	147	179	267
2019	92	122	201
2020	42	81	184
2021	103	148	245
2022	92	126	242

## Data Availability

The data are the property of the Pollutrack company. Some data can be requested to the Pollutrack company in the frame of scientific research projects.

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
