# Peer review of "Spatial Distribution of PM2.5 Mass and Number Concentrations in Paris (France) from the Pollutrack Network of Mobile Sensors during 2018–2022"

_sensors, 2023, doi:10.3390/s23208560_

Round 1

Reviewer 1 Report

Major comments

In my opinion, the manuscript as presented cannot be published. The entire manuscript should be rewritten. The Authors should describe the methodology and the results and conclusions in such a way that they are understandable to a wide audience.

I encourage you to be concise in your text. All introductory information justifying the research, the choice of research goals and scope, as well as the choice of methods should be described in one, possibly concise, Introduction chapter. I advise the Authors of the manuscript to read the Instructions for Authors of the Sensors magazine and prepare the article in accordance with the recommendations, in particular the Research Manuscript Sections.

Detailed comments

1.      Introduction

v. 43; nitrous oxide and sulfur oxide - I think you meant nitrous oxides and sulfur oxides.

v. 44; “mainly attributed to anthropogenic sources (secondary PM) - ”anthropogenic pollutants” and “secondary pollutants” do not mean the same.

At the end of the Introduction chapter, the purpose and scope of the research are given in a very imprecise manner. The purpose and scope of the research should be more precisely justified and described.

2.2. Validation of the sensors

This subsection contains the results of intercalibration experiments, but the experiment itself is not described: it is not known how long the experiment lasted, under what conditions it was carried out, how many sensors were involved, what sensors, how the concentrations were averaged for comparison, how the results were compared, etc. It is not known what show the figures and tables in this subsection. Where does the conclusion come from that the measurement uncertainty is 3 μg.m-3? There are too many ambiguities in this part of the work.

If PM2.5 is in the title, all measurements should refer to PM2.5.

3.2. Research methods

Unfortunately, there is no such chapter in the manuscript. This is inconsistent with the principles of scientific publishing and the recommendations of the Sensors journal.

3. PM2.5 Maps

The article presents charts and maps with the results, but does not describe what they show or how they were created. How often measurements were taken in  specific areas.

5. Health consequences

The content of section 5 is inconsistent with the title Health consequences. The authors refer to two references ([33] and [14]), which either do not concern Paris or do not concern the period of the experiment. There are no legitimate literature references in this section. This fragment of the work does not bring anything new to the state of knowledge.

6. Conclusions

“When compared to mass-concentrations, the results in number concentrations of PM seem to be better adapted to evaluate what the citizens really inhale daily in terms of deleterious particles with a pejorative health impact, an information that regulatory monitoring based on PM10 weighing cannot accurately produce. The following parts of the text are also unclear.”

English language needs improvement. There are also errors in the construction of entire sentences that make it very difficult to understand the meaning of the sentence.  There are also many terminological errors.

Author Response

REVIEWER 1

Major comments

In my opinion, the manuscript as presented cannot be published. The entire manuscript should be rewritten. The Authors should describe the methodology and the results and conclusions in such a way that they are understandable to a wide audience.

I encourage you to be concise in your text. All introductory information justifying the research, the choice of research goals and scope, as well as the choice of methods should be described in one, possibly concise, Introduction chapter. I advise the Authors of the manuscript to read the Instructions for Authors of the Sensors magazine and prepare the article in accordance with the recommendations, in particular the Research Manuscript Sections.

Authors answer: We thank the reviewer for their pertinent comments. Nevertheless, we don’t well understand what the reviewer means by “to be concise”. The paper is short (340 lines), thus it seems difficult to be more concise. This comment seems to be partly in contradiction with the following comments asking to provide more descriptions and elements on data analysis. Finally, it seems to us that all scientific papers don't always need to have same scholastic structure. Some experiments and analysis should need specific sections. The recommendations for Sensors are recommendations, not obligations.

We have tried to do our best to improve the paper following the reviewer comments.

Detailed comments

  1. Introduction
  2. 43; nitrous oxide and sulfur oxide - I think you meant nitrous oxides and sulfur oxides.

            Authors answer: Correction done.

  1. 44; “mainly attributed to anthropogenic sources (secondary PM) - ”anthropogenic pollutants” and “secondary pollutants” do not mean the same.

Authors answer: Indeed, there is a confusion in our text. We have changed it to: “these particles can be directly produced by various anthropogenic sources, mainly traffic, building activities, wood and oil-fired boiler heating (primary PM), but also from complex chemical reactions involving sunlight, anthropogenic ammonium, nitrous oxides and sulfur oxides (secondary PM).”

At the end of the Introduction chapter, the purpose and scope of the research are given in a very imprecise manner. The purpose and scope of the research should be more precisely justified and described.

Authors answer: The description of the work was partly given in the last sentence of this part. We have changed the text to: “The aim of the present paper is to go further in the analysis of the mean spatial variability of PM2.5 across Paris and its motorway ring, over the 2018-2022 period, using 1-km2 resolution maps for the PM2.5 mass-concentrations levels and for number of days above the WHO recommendations. We will present the measurement strategy, the instrument accuracy determined from comparison with reference measurements, the temporal variability, the 1-km2 resolution maps of the spatial variability, the corresponding number concentrations, and we will discuss the meaning of the results in terms of human health.”

2.2. Validation of the sensors

This subsection contains the results of intercalibration experiments, but the experiment itself is not described: it is not known how long the experiment lasted, under what conditions it was carried out, how many sensors were involved, what sensors, how the concentrations were averaged for comparison, how the results were compared, etc. It is not known what show the figures and tables in this subsection. Where does the conclusion come from that the measurement uncertainty is 3 μg.m-3? There are too many ambiguities in this part of the work.

If PM2.5 is in the title, all measurements should refer to PM2.5.

Authors answer: The reviewer is correct; some information is missing concerning the duration of the inter-comparisons sessions. We have added in the text for the Paris region: “traffic condition in the motorway ring (“Peripherique Est”) from 26 April to 15 October 2022, urban background (garden in the center of Paris “Les Halles”) from 13 July to 15 October 2022, and rural condition (“Rambouillet”) from 10 July to 15 October 2022.” For the Dublin session, we have added “for the entire year 2022”.

The name of the reference instruments for Paris are now given (BAM 1020 from Metone). For the Dublin sessions we have changed the text to: “To confirm these results, three other sessions of intercomparison with Pollutrack sensors and PM2.5 reference microbalance were conducted at 3 different locations in Dublin (Ireland), which is a port city, for the entire year 2022. Similar results are obtained for the measurements statistics (Table 2) than for the Paris region”.

We have changed the text to: “The daily difference between the Pollutrack and reference measurements are calculated, then the mean difference is calculated. For all these inter-comparison sessions, a mean value of ~3 µg.m-3 is found. It can be concluded that the values of individual Pollutrack sensors have a mean uncertainty of 3 µg.m-3.”

The number of sensors (Pollutrack and reference instruments) have been already given in the text. Also, it is already said that the data are daily averaged. And indeed, all these measurements refer to PM2.5 (several “PM2.5” have been added in the text when necessary).

3.2. Research methods

Unfortunately, there is no such chapter in the manuscript. This is inconsistent with the principles of scientific publishing and the recommendations of the Sensors journal.

Authors answer

The information is given in part 2.1 (Strategy of measurements) and in part 3.1 (Maps retrievals). Other information concerning the sensors is available in the previous Pollutrack paper cited frequently in the text.

  1. PM2.5 Maps

The article presents charts and maps with the results, but does not describe what they show or how they were created. How often measurements were taken in  specific areas.

Authors answer: Indeed, some information was missing and our text was confusing. The 3.1 part is totally rewritten: “The total surface of Paris city is divided into squares of 1-km side length to ensure hundreds of Pollutrack measurements per day and per square. Then, these data are integrated over 1 day, to provide daily values per square for the 5-years period. This is statistically relevant since the measurements are randomly distributed along the daytime and since the cars daily cover all of the Paris main streets. Such 108 parcels provide a strong improvement in spatial sampling compared to the 3 to 6 PM2.5 stations of the Airparif air quality monitoring network. It is then possible to study the temporal trend of the PM2.5 pollutants per 1 km2 square.

Renard et al. [22] has used a different approach using the 2020 Pollutrack data, to produce examples of daily PM2.5 maps with a resolution of 100 m along main roads only. This approach is specifically useful to detect hot spots but is less adapted to point out the spatial global trend. Thus, both approaches are complementary.

The 108 Pollutrack daily values are averaged to produce one global value per day that can be compared to the mean of the daily Airparif fixed station values, to verify the consistency of the two sets of data. As shown in the correlation plot of Figure 3, the agreement between the two sets is very good, with a mean difference of 0.1±3.5 µg.m-3. It can be concluded that first the mean of the few fixed monitoring stations can provide a good estimate of the global pollution in Paris, except for a few measurements where the values are too high due to local and short-duration pollution events as the isolated 70 µg.m-3 Airparif value in the Figure 2, and secondly that no systematic bias is present in the Pollutrack measurements.

Finally, the data can be also averaged for each 1 km2 square over the 5-years period of measurements to establish the mean spatial trend of the PM2.5 pollution levels in Paris.”

The title of part 3.3 is changed to: “Spatial trend”.

  1. Health consequences

The content of section 5 is inconsistent with the title Health consequences. The authors refer to two references ([33] and [14]), which either do not concern Paris or do not concern the period of the experiment. There are no legitimate literature references in this section. This fragment of the work does not bring anything new to the state of knowledge.

Authors answer: We have already given three references. But indeed, this part needed some improvements. We have changed the title to: 4.3. Towards a better analysis of the consequence of the PM pollution on health “ and the text to: “The high level of PM2.5 pollution can explain the high mortality that occurred in Paris during the Covid-19 pandemic, one of the highest values in western Europe [10,11]. It would have been of strong interest to tentatively correlate the spatial trend of Covid-19 mortality with the spatial trend of PM2.5 pollution inside Paris, but unfortunately, such medical data are not currently available. Similarly, to better understand the effect of PM2.5 on severe respiratory pathologies, future analysis will be necessary using mortality data in parallel with PM2.5 data, but again such data are not currently available for that purpose.

The analysis presented here should be conducted for other polluted cities in the world. Most of the overall studies (~ 69%) on the health effect of PM pollution were carried out in the cities of high-income countries despite the fact that PM2.5 concentrations are higher in the Low and Middle Income Countries [35] where severe sanitary effects should be expected. Finally, chemical components and sources apportionment of PM2.5 to the associated toxicity is poorly documented. Thus, a better determination of PM origins should be conducted in parallel with the PM content and health analysis, using instruments which can provide typology and composition of the PM (as done during pollution events in Paris [14]), and the access to more local sanitary data.”

  1. Conclusions

“When compared to mass-concentrations, the results in number concentrations of PM seem to be better adapted to evaluate what the citizens really inhale daily in terms of deleterious particles with a pejorative health impact, an information that regulatory monitoring based on PM10 weighing cannot accurately produce”. The following parts of the text are also unclear.

 Authors answer: We have changed the text to: “When compared to regulatory monitoring based on PM10 weighing mass-concentrations (and even PM2.5), the results in number concentrations of PM seem to be better adapted to evaluate what the citizens really inhale in terms of deleterious particles with a pejorative health impact. The number of particles per cm3 increase when the particle size decreases, and the maximum number concentration is at around 100 nm for urban pollution. Nevertheless, the mass-concentrations of such submicron particles remain weak when compared to the largest ones, and thus do not always strongly contribute to the final PM10 and PM2.5 mass-concentrations. Thus, the amount of the smallest particles cannot be accurately determined when using the mass-concentrations instruments, and pollution events dominated by such dangerous tiny particles could be undetected when considered only the regulatory pollution thresholds.

Comments on the Quality of English Language. English language needs improvement. There are also errors in the construction of entire sentences that make it very difficult to understand the meaning of the sentence.  There are also many terminological errors.

Authors answer: The paper is now reviewed by an English native.

Reviewer 2 Report

I see no problem with this work. The authors conducted a high-level research. This study is currently relevant and touches on many important issues.

I also liked that the authors focused on the most relevant problem of measuring and assessing air pollution levels in the local areas of the Cities. In particular, it is discussed that measuring mass concentration is an outdated approach and does not reflect the whole picture. Measuring number concentration may be more indicative. The relevance of measuring submicron particles for air pollution analysis is also discussed. However, it should be noted that this study has limitations. The sensors used have a limit of 500 nm, while pollution with submicron particles is within 100-1000 nm and often has a median of 300-500 nm. As a result, assessing air pollution using such sensors is always associated with some degree of error and underestimation of submicron particle indicators. I consider it important to note this in the "limitations of the study" section.

Nevertheless, this remark does not diminish the importance of the study.

Notes: Underlined and highlighted text should not be present in the document. There is no reference to WHO in lines 44-46. There is a spelling error in line 267: "Some differences can be pointed out between...".

Author Response

     REVIEWER 2

I see no problem with this work. The authors conducted a high-level research. This study is currently relevant and touches on many important issues.

I also liked that the authors focused on the most relevant problem of measuring and assessing air pollution levels in the local areas of the Cities. In particular, it is discussed that measuring mass-concentration is an outdated approach and does not reflect the whole picture. Measuring number concentration may be more indicative. The relevance of measuring submicron particles for air pollution analysis is also discussed. However, it should be noted that this study has limitations. The sensors used have a limit of 500 nm, while pollution with submicron particles is within 100-1000 nm and often has a median of 300-500 nm. As a result, assessing air pollution using such sensors is always associated with some degree of error and underestimation of submicron particle indicators. I consider it important to note this in the "limitations of the study" section.

Nevertheless, this remark does not diminish the importance of the study.

Authors answer: “We thank the reviewer for their comments. We have added in the text: Finally, conventional optical counters cannot detect particles smaller than about 200 nm, although the maximum concentration of urban pollution particles is at around 100 nm for urban pollution. Thus they underestimate the number of submicron particles, although providing an acceptable indicator of their presence. “

Notes: Underlined and highlighted text should not be present in the document.

Authors answer: These errors are removed.

There is no reference to WHO in lines 44-46.

Authors answer: The reference is added.

There is a spelling error in line 267: "Some differences can be pointed out between...".

Authors answer: Corrected.

Reviewer 3 Report

Comments to the authors regarding MS entitled “Spatial distribution of PM2.5 mass and number concentrations in Paris (France) from the Pollutrack network of mobile sensors in the 2018-2022 period”.

Line 40: should be ≤ 2.5 µm

Line 64-71: References to support these statements.

Line 83: could the authors provide implications extracted from this study?

Lines 91-95: Could the sensor be affected by wind? Please clarify.

Lines 114-117: A map showing grid distribution is suggested.

From Tables 1 and 2, the slopes indicated that the bias between the 2 methods is about 20%, please provide an explanation for that. Please also add the data number for these comparisons.

Lines 168-169: odd term “global pollution”. More details about the integration method of data need to be provided in this section.

Fig. 3: The data is more deviate when PM2.5 > 30 ug/m3, any possibility?

Line 176: Odd term “The high season of PM pollution”

Lines 182: odd term “global trend”, please be careful to use the term global since it does not make any sense in the concept of this study.

Table 3: Why lowest/highest mean during 2018-2022 different from the lowest/highest for every year? Similar question for Table 4.

Lines 196-203: Not a good way to present in scientific paper. What is “ring”?  

I would suggest merging the results + discussion sections to improve the clarification.  

Discussion regarding major driving factors of PM2.5 pollution in this study is relatively subjective without good supporting information and evidence. For example, Lines 178-179; Lines 184-186; 209-210 etc. Even in the discussion section, it is also unclear.   

Carefully check the typo throughout the MS (e.g., Line 267)

I would suggest the authors improve the quality of Fig. 2-8 due to their very low resolution. In addition, please use other software (e.g., Q-GIS) to re-plot Fig. 5, 6, 8. The recent plots are not a good way to illustrate spatial distribution in terms of hotspot detection. Also, the caption of Fig. 8 is incorrect.  

Author Response

REVIEWER 3

Comments to the authors regarding MS entitled “Spatial distribution of PM2.5 mass and number concentrations in Paris (France) from the Pollutrack network of mobile sensors in the 2018-2022 period”.

Line 40: should be ≤ 2.5 µm

Authors answer: We thank the reviewer for their comments. Correction done.

Line 64-71: References to support these statements.

Authors answer: A reference is added.

Line 83: could the authors provide implications extracted from this study?

Authors answer: Some implications are discussed in the (rewritten) part 4.3.

Lines 91-95: Could the sensor be affected by wind? Please clarify.

Authors answer: The reviewer raises an interesting point. The instruments have been calibrated to be used with relative wind up to around 40 km.h-1, as said in the text. In case of low wind, the pollution is high, and it has been verified that the sensors work well. In case of strong winds, the pollution will be (very) low and a potential effect of the winds on the measurements will be insignificant.

Lines 114-117: A map showing grid distribution is suggested.

Authors answer: The grid is shown in Figures 5, 6 and 8. It seems unnecessary to plot the grid before.

From Tables 1 and 2, the slopes indicated that the bias between the 2 methods is about 20%, please provide an explanation for that. Please also add the data number for these comparisons.

Authors answer: It is not two methods, but the same method applied to two different locations (Paris and Dublin). We think that this 20% value is coming when considering the slopes on the correlation plots. The slopes can be influenced by few isolated values, as some of the highest mass-concentration values. If all the data are plotted all together, the value of the slope will be very close to 1. It is the reason why other parameters must be considered: the value at origin, and the standard deviation.

Since several hundred of daily data are used, it seems unnecessary to provide them in a table.

We have added in the text: “and the presence of few isolated values for the highest mass-concentrations can bias the retrieval of the slope values on the correlation plots.”

Lines 168-169: odd term “global pollution”. More details about the integration method of data need to be provided in this section.

Authors answer: We have changed “global” to “mean”.

The part 3.1 is totally rewritten: “The total surface of Paris city is divided into squares of 1-km side length to ensure hundreds of Pollutrack measurements per day and per square. Then, these data are integrated over 1 day, to provide daily values per square for the 5-years period. This is statistically relevant since the measurements are randomly distributed along the daytime and since the cars cover all of the Paris main streets. Such 108 parcels provide a strong improvement in spatial sampling compared to the 3 to 6 PM2.5 stations of the Airparif air quality monitoring network. It is then possible to study the temporal trend of the PM2.5 pollutants per 1 km2 square.

Renard et al. [22] has used a different approach using the 2020 Pollutrack data, to produce examples of daily PM2.5 maps with a resolution of 100 m along main roads only. This approach is specifically useful to detect hot spots but is less adapted to point out the spatial global trend. Thus, both approaches are complementary.

The 108 Pollutrack daily values are averaged to produce one global value per day that can be compared to the mean of the daily Airparif fixed station values, to verify the consistency of the two sets of data. As shown in the correlation plot of Figure 3, the agreement between the two sets is very good, with a mean difference of 0.1±3.5 µg.m-3. It can be concluded that first the mean of the few fixed monitoring stations can provide a good estimate of the mean pollution in Paris, expect for a few measurements where the values are too high due to local and short-duration pollution events as the isolated 70 µg.m-3 Airparif value in the Figure 2, and secondly that no systematic bias is present in the Pollutrack measurements.

Finally, the data can be also averaged for each 1 km2 square over the 5-years period of measurements to establish the mean spatial trend of the PM2.5 pollution levels in Paris.”

Fig. 3: The data is more deviate when PM2.5 > 30 ug/m3, any possibility?

Authors answer: It is difficult to conclude this, since the number of measurements is too low. We agree that this fact is present for the example presented here, but does not occur for most of the validation sessions.

Line 176: Odd term “The high season of PM pollution”

Authors answer: We have added in the text: “The “high season” of PM pollution (meaning the period of the year where the highest mass-concentrations values are measured)”

Lines 182: odd term “global trend”, please be careful to use the term global since it does not make any sense in the concept of this study.

Authors answer: We have removed the word “global”.

Table 3: Why lowest/highest mean during 2018-2022 different from the lowest/highest for every year? Similar question for Table 4.

Authors answer: We have changed the legend to: “Lowest yearly mean, yearly mean and highest yearly mean mass-concentration values in the 108 1-km2 squares.”

Lines 196-203: Not a good way to present in scientific paper. What is “ring”?  

Authors answer: We have replaced all “ring” by “motorway ring”.

I would suggest merging the results + discussion sections to improve the clarification.  

Discussion regarding major driving factors of PM2.5 pollution in this study is relatively subjective without good supporting information and evidence. For example, Lines 178-179; Lines 184-186; 209-210 etc. Even in the discussion section, it is also unclear.   

Authors answer: The discussion part is long (now the previous section 5 is merged with the section 4), thus it seems difficult to merge it with the results section (PM2.5 maps).

The discussion is not relatively subjective, since the driving factors for the Paris pollution are well documented. Two references were given in previous lines 178-179. One reference is given in lines 184-186. For previous lines 209-210, concerning the contribution of the motorway ring we have added: “At distance larger than about 1 km, the effect of the motorway pollution seems to disappear.”

Finally, since the discussion covers different topics, we don’t understand which subsection is unclear.

Carefully check the typo throughout the MS (e.g., Line 267)

Authors answer: Correction done.

I would suggest the authors improve the quality of Fig. 2-8 due to their very low resolution. In addition, please use other software (e.g., Q-GIS) to re-plot Fig. 5, 6, 8. The recent plots are not a good way to illustrate spatial distribution in terms of hotspot detection. Also, the caption of Fig. 8 is incorrect.  

Authors answer: In the original word file, the quality of the figure seems to be correct.

The goal of the paper is not to detect hotspot but to reveal the spatial variation. We don’t see another way to illustrate this phenomenon.

Indeed the caption of Figure 8 and is now corrected to: “number concentration”

Reviewer 4 Report

The authors present the results from 3 years of measurements of PM2.5 using Pollutrack sensors. This is an interest project and very useful tool to monitor air quality. 

I just have a few comments, I think the article needs some english revision. The quality of the graphs are not very good, don't know if this is because loosing quality when generating the PDF, but I recommend to improve them. Also in Figure 2, you need to check the units for the air concentration, they are different (ug/m3 and ug.m-3).

There are also some typos, like in lines 37-38, 51-54 and 56-60

It needs some english revision

Author Response

REVIEWER 4

The authors present the results from 3 years of measurements of PM2.5 using Pollutrack sensors. This is an interesting project and very useful tool to monitor air quality. 

I just have a few comments, I think the article needs some English revision.

Authors answer: The paper is now corrected by an English native.

The quality of the graphs are not very good, don't know if this is because loosing quality when generating the PDF, but I recommend to improve them.

Authors answer: The quality of the original figures seems to be correct in the word file.

Also in Figure 2, you need to check the units for the air concentration, they are different (ug/m3 and ug.m-3).

Authors answer: The legend is corrected.

There are also some typos, like in lines 37-38, 51-54 and 56-60

Authors answer: Corrections done.

Round 2

Reviewer 1 Report

I accept the new version of the article. However, I believe that the text of the manuscript is still written in a language that is not very communicative. It will certainly be difficult to read for a non-specialist. I hope that the editorial correction will alleviate this inconvenience to some extent.

I think that the authors could work on the precision of the text and its communicativeness.

Reviewer 3 Report

No more comments. For the quality, I think can consider publishing in Sensor journal.